

# Structural equation models of health behaviour, psychological well-being, symptom severity and quality of life in abdominal bloating

Nurzulaikha Abdullah[1,2], Yee Cheng Kueh[1], Garry Kuan[3], Mung Seong Wong[4], Vincent Tee[4], Tengku Ahmad Iskandar Tengku Alang[4], Nurhazwani Hamid[4] and Yeong Yeh Lee[4,5]

[1] Biostatistics and Research Methodology Unit, School of Medical Sciences, Universiti Sains Malaysia, Kubang Kerian, Kelantan, Malaysia
[2] Faculty of Data Science and Computing, Universiti Malaysia Kelantan, Kota Bharu, Kelantan, Malaysia
[3] Exercise and Sport Science, School of Health Sciences, Universiti Sains Malaysia, Kelantan, Kubang Kerian, Malaysia
[4] Department of Medicine, School of Medical Sciences, Universiti Sains Malaysia, Kubang Kerian, Kelantan, Malaysia
[5] GI & Motility Unit, Hospital Universiti Sains Malaysia, Kubang Kerian, Kelantan, Malaysia

Corresponding author
Yee Cheng Kueh, yckueh@usm.my

## ABSTRACT

**Background:** The objective of this study was to investigate the inter-relationship between psychosocial variables and their impact on symptom severity and quality of life (QoL) concerning abdominal bloating.

**Methods:** The study adopted a cross-sectional design with purposive sampling. Participants who consented and met the criteria for bloating based on the Rome IV classification completed designated questionnaires. Independent variables comprised health beliefs, intentions, health-promoting behaviors, social support, depression, and anxiety, while dependent variables included bloating severity (general and within 24 h) and QoL. Structural Equation Modeling (SEM) was conducted utilizing Mplus 8.0 to analyze the relationships between these factors.

**Results:** A total of 323 participants, with a mean age of 27.69 years (SD = 11.50), predominantly females (64.7%), volunteered to participate in the study. The final SEM model exhibited good fit based on various indices (CFI = 0.922, SRMR = 0.064, RMSEA (95% CI) = 0.048 (0.041–0.054), $p$-value = 0.714), with 15 significant path relationships identified. The model explained 12.0% of the variance in severity within 24 h, 6% in general severity, and 53.8% in QoL.

**Conclusion:** The findings underscore the significant influence of health beliefs, intentions, behaviors, social support, depression, and anxiety on symptom severity and QoL in individuals experiencing abdominal bloating.

## INTRODUCTION

The symptoms of abdominal bloating (AB) are prevalent in the general population, often experienced without seeking medical treatment. Study has shown that AB can significantly diminish the quality of life (QoL), a phenomenon that is increasingly recognised. Malaysians, influenced by the diverse culinary offering resulting from the fusion of carious cultural traditions, are more prone to experiencing AB (*Abdullah et al., 2021b*). According to *Malagelada, Accarino & Azpiroz (2017)*, AB was one of the common symptoms in a healthy adult and reduced the QoL by negatively impacting the adult's well-being. Besides, *Lacy & Patel (2017)* reported that more than half of the irritable bowel syndrome (IBS) patients complained about reducing daily activities due to AB. AB was believed to be one of the contributors to the severity of IBS, which led to an increase in health care utilisation (*Lacy & Patel, 2017*). Symptoms of AB pose a disturbance in almost any individual's busy daily routine as it dramatically impacts one's well-being. In addition, it might also trigger other problems and deteriorate health, thus it should not be taken lightly. Yet, the researchers found a lack of research on the related variables among Malaysian people with AB.

Even though AB symptoms were not life threatening, several researchers reported it as a cause for the limitation in daily activities (*Kanazawa et al., 2016*). It can increase absenteeism, reduce productivity, increase healthcare costs, and reduce the chances of elongating years of living. AB was also a significant marker for various underlying diseases (*Tuteja et al., 2008*). The most common related disorders associated with AB were gastrointestinal problems such as IBS, irritable bowel disease (IBD), colon cancer, and stomach cancer. Physicians may perceive AB as common and bothersome but not enough to be life-threatening. Thus, AB is less frequently explored despite its prevalence and importance. Due to the above reasons, researchers have shown a growing interest in the treatment of AB to reduce the frequency of episodes or its severity (*Thiwan, 2019*; *Kamboj & Oxentenko, 2018*; *Khoshoo, Armstead & Landry, 2006*).

The Theory of Planned Behaviour (TPB) proposed that belief, intention, and behaviour correlate to each other and manipulating these three cores concept may help to improve a symptom or disease (*Ajzen, 2012*). Initially developed in 1990 by Ajzen, the TPB evolved through the ideas which circled around the prediction of behaviour and extended originally from the Theory of Reasoned Action (TRA) with the additional domain of perceived behaviour control under the health belief construct (*Ajzen, 2012*). There were many research associate TPB with other different and variety of variables (*Bai & Dinour, 2017*; *Chin & Mansori, 2019*; *Li, Figg & Schüz, 2019*).

The recent treatment algorithm has promoted the implementation of a Multidimensional Clinical Profile (MDCP) and the Biopsychosocial model to provide better patient management (*Drossman, 2017*; *Levy et al., 2006*; *Oudenhove et al., 2016*; *Schmulson & Drossman, 2017*). Both models emphasized the psychological comorbidities, impacts, and disturbances aside from the contemporary clinical kaleidoscope. Therefore, understanding the psychological and psychosocial variables among patients with AB is crucial in treating patients from the disorder of gut-brain interaction (DGBI) spectrum, especially IBS.

The conceptual framework of the current study is founded on these models, incorporating a comprehensive range of variables to better comprehend outcomes, not confined solely to a clinical standpoint. *Simões et al. (2016)* proposed that individuals can enhance their quality of life (QoL) through improved adaptive behavior skills, even with less social support. In contrast, *Dogan & Tan (2019)* highlighted the pivotal role of social support in fostering good QoL. Additionally, health-promoting behaviors play a crucial role in health education, commencing from early stages (*Kuan et al., 2019*). *Kuan (2023)* discovered that increased physical and mental activity correlates with improved mental health due to heightened life satisfaction and reduced psychological distress. Depression and anxiety, considered as indicators of psychological distress alongside stress and life satisfaction, underscore the multifaceted nature of mental health. Given that mental health is influenced by various factors such as personality traits, living conditions, and major life events affecting QoL satisfaction, proactive lifestyle modifications are crucial in curbing the prevalence of psychological disorders. Understanding the levels of social support, depression, and anxiety among individuals with AB can contribute to a deeper understanding of related variables that influence QoL in this specific cohort.

Some literature suggests that IBS significantly lowers the QoL and affects their activities of daily living, which are commonly present in people with IBS and other diseases (*Kanazawa et al., 2016*; *Thiwan, 2019*; *Kamboj & Oxentenko, 2018*). By reducing the severity of AB, the QoL and productivity of individual with AB is expected to be improved. Therefore, more research has been initiated to explore the alternative ways to treat AB, considering various aspects beyond just the clinical perspective, but also integrating cultural and environmental influences. For nearly two decades, other than medication, other psychological or social variables have been considered as foundation in the AB management (*Drossman, 2006*; *Tanaka et al., 2011*; *Cao et al., 2020*; *Chin & Mansori, 2019*; *Bai & Dinour, 2017*; *Li, Figg & Schüz, 2019*). Developing and maintaining positive attitudes and psychological behaviour toward addressing AB is crucial, as it may be an approach for the management and treatment of the disease. By taking into account mediating factors, individuals with AB can potentially mitigate or exacerbate the impact of the condition on outcomes such as severity and QoL through the management of other variables such as intention, social support, depression, and anxiety. The application and integration of these constructs among Malaysians with AB could potentially have a positive impact on their symptom management. Therefore, the primary objective of this study is to investigate the direct and indirect relationships between TPB psychological constructs, other psychosocial variables, symptom severity, and QoL among Malaysians living with AB, utilizing structural equation modeling (SEM) as the analytical approach.

By examining these interrelated factors, the study seeks to provide insights into how psychological factors and psychosocial variables influence symptom management and overall quality of life for individuals dealing with AB in Malaysia. This research aims to contribute valuable knowledge that could inform interventions and strategies aimed at improving the well-being and health outcomes of individuals with AB in the Malaysian context.

Figure 1 shows the conceptual framework proposed in the present study.
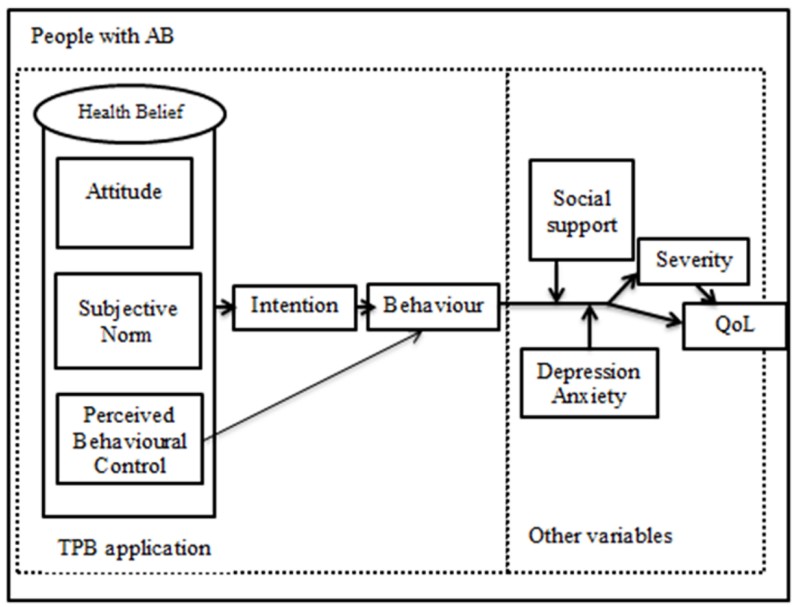

**Figure 1 Hypothesised conceptual framework.**

## MATERIALS & METHODS

### Study design, ethical approval

A cross-sectional study design was employed. People with AB who satisfied the Rome IV criteria for AB were recruited *via* purposive sampling from Hospital Universiti Sains Malaysia, the largest tertiary center in the north-eastern region of Peninsular Malaysia. Briefly, the Rome IV criteria (*Mari et al., 2019*) for AB are as follow: (1) recurrent feeling of AB or visible distention for at least 1 day per week, (2) onset of symptoms at least 6 months before diagnosis, (3) the presence of symptoms for at least 3 months and insufficient criteria to establish other diagnosis and (4) may also co-exist with mild abdominal pain and minor bowel disorders. All participants who met the eligibility criteria of the study, including the following: ROME IV criteria, aged 18 years and above, in the absence of organic GI diseases (*e.g.*, inflammatory bowel disease, gastrointestinal (GI) infections and colorectal cancer), no past abdominal surgery, not on drugs which cause or worsen AB (*e. g.*, opiates), and did not have any psychiatric illnesses (*e.g.*, schizophrenia) were approached. All participants were required to complete an informed consent form before being recruited. Ethical approval from the Human Research Ethics Committee of Universiti Sains Malaysia (USM/JEPeM/17010012)) was obtained. The study was conducted in accordance with the Declaration of Helsinki.

### Instruments

#### Demographic information

Socio-demographic information included the following: age, sex, body mass index (BMI), living area (either rural or urban) and history of other symptoms.

### Health belief for bloating (HB-Bloat) scale

Based on the TPB, the Malay version of the health belief for bloating (HB-Bloat) scale is a 32-items initially developed by *Abdullah et al. (2020)* measuring the health belief or individuals' perception toward AB improvement. The scale is divided into three main subscales, including attitudes (13 items), subjective norm (eight items) and perceived control (11 items). The items were measured on 5-point Likert scales ranging from 1 (strongly agree) to 5 (strongly disagree), where higher scores reflect positive belief toward AB improvement. Model fit indices of HB-Bloat also showed an acceptable range value of comparative fit index (CFI) = 0.941, Tucker-Lewis index (TLI) = 0.924, root mean square error of approximation (RMSEA) = 0.054, and standardized root mean square residual (SRMR) = 0.044. HB-Bloat also showed acceptable reliability based on composite reliability, 0.70 for attitude, 0.78 for subjective norm and 0.75 for perceived behavioural control.

### Intention

One item measures the intention to promote healthy behaviour towards AB improvement. It consisted of 5-point Likert scales ranging from 1 (strongly agree) to 5 (strongly disagree), where a higher score reflects a greater intention to improve AB.

### Health promoting behaviour for bloating (HPB-Bloat) scale

The HPB-Bloat scale consists of 35 items with five subscales. The scale measures individuals' daily lifestyles and behaviour that are helping them reduce AB symptoms (*Abdullah et al., 2021b*). The domains include any activity undertaken by a person believing him or herself to be healthy for improving AB or preventing it from deteriorating stage, and a person doing it either through (1) diet, (2) health awareness, (3) stress management, (4) physical activity or (5) alternative treatments. Responses are 5-point Likert scales ranging from 1 (never) to 5 (very often), where higher scores reflect a greater lifestyle change to improve AB. The HPB-Bloat demonstrated good model fit indices, with CFI = 0.93, TLI = 0.91, RMSEA = 0.044, and SRMR = 0.052. The composite reliability ranged from 0.64–0.82.

### Social support for bloating (SS-Bloat) scale

Social support refers to any support given by any individual, either their spouse or other closest relatives or friends, in dealing with AB symptoms either physically or emotionally. Responses are on a 5-point Likert scale ranging from 1 (strongly agree) to 5 (strongly disagree), where higher scores reflect greater social support. This scale has five items under one subscale (*Abdullah et al., 2021c*). The total variance explained by the EFA model was 35.6%, and the Cronbach alpha of the single factor was 0.66.

### Hospital Anxiety and Depression Scale (HADS)

The Malay version of HADS has been validated by *Yahaya & Othman (2015)*. The 14-item HADS is used to assess the presence of psychological disorders, including depression and anxiety (*Yahaya & Othman, 2015*) in the general medical population of patient (*Stern, 2014*). Eventhough it was devised 30 years ago by the original author of

*Zigmond & Snaith (1983)*. HADS is one of the ways to discover the symptoms of anxiety or/and depression (none to severe cases), even though it is merely a non-physical symptom it can trigger more problem if treated lightly. So, we want to cater all level of anxiety and depression which we believe can be trace using HADS. It was reported that HADS does not include all the diagnostic criteria of depression where additional question on appetite, sleep and self/harm/suicidal though have to be asked for risk assessment, we believe that in the future study it can be additional point to be noted. The scores can be summed up for each subscale of anxiety and depression or summed up as a total score to represent the level of psychological distress (*Montazeri et al., 2003*; *Snaith, 2003*). HADS is well-established and widely used (*Azlan et al., 2020*; *Basnayake et al., 2020*; *Khan et al., 2019*; *Tominaga et al., 2018*; *Yousuf et al., 2020*). The sensitivity and specificity values for depression were 93.2% and 90.8%, respectively, and for anxiety, 90.0% and 86.2%, respectively (*Yahaya & Othman, 2015*).

### Malay version of the Bloating Severity Questionnaire (BSQ-M)

The original English version of the 12-item BSQ was developed to measure the symptom severity of AB (*Palsson et al., 2004*; *Thiwan et al., 2005*; *Thiwan, Whitehead & Palsson, 2004*). It measures overall severity (general or SevGen, 7-item) and severity for the past 24 h (Sev24, 5-item). Responses for BSQ are in the format of multiple-choice answers on a different degree of effect on individuals based on intensity, frequency, and severity (less severe to more severe, ranging from 1–4 or 5–6 or 7 or 8 varied by items). The present study used the translated Malay version BSQ-M (*Mahd-Ab.lah et al., 2021*), which has been validated with good fit indices (CFI = 0.976, TLI = 0.956, RMSEA = 0.050 and SRMR = 0.051). For SevGen, items 1, 3, 4, 5, and 6 measured 1–5 scale, item 2 measured 1–4 scale, and item 7 measured 1–7 scale. For sev24, all items measured 1–5 scale except for item 5 (estimated 1–8 scale). Both subscales of BSQ had good internal consistency (Cronbach alpha values for SevGen and Sev24 were 0.76 and 0.85, respectively (*Palsson et al., 2004*). There is an abbreviated 3-item Sev24, with similar internal consistency to 5-item Sev24, but the original 5-items Sev24 was used (*Palsson et al., 2004*; *Thiwan et al., 2005*; *Thiwan, Whitehead & Palsson, 2004*). The composite reliability of the two domains, *i.e.*, SevGen and Sev24 were 0.797 and 0.909, respectively.

### Malay version of Bloating Quality of Life (BLQoL-M)

The original 5-item BLQoL was developed by *Palsson et al. (2004)* to measure the impact of AB on QoL. The BLQoL included questions on interference with work, intimate relationships, hobbies, social activities, and emotion. Responses are on seven points Likert scales ranging from 1 = "never/not related to me" to 7 = "always" with higher scores indicating a greater impact on QoL (*Palsson et al., 2004*). The present study used the translated and validated Malay version of BLQoL-M (*Mahd-Ab.lah et al., 2021*) with good fit indices (CFI = 0.955, TLI = 0.962, RMSEA = 0.071 and SRMR = 0.021).

## Procedure

The study was conducted around Hospital Universiti Sains Malaysia, Kelantan, Malaysia where purposive sampling method was employed.

All possible participants were further screened according to the inclusion and exclusion criteria. Written consent was obtained before inclusion in the study. The present study used the self-reported questionnaires with additional sociodemographic questions. The participants voluntarily completed the questionnaires and returned it to the researchers. The estimated time to complete the survey was 40–45 min.

There were 355 new potential participants screened and eventually, 330 participants ful-filled the eligibility criteria and were invited to complete the survey. Among all who re-turned the questionnaires, 323 were complete and usable for the subsequent SEM data analysis. The response rate was 97.9% which was considered good.

## Data analysis

Mplus 8.0 was used to perform the statistical analysis. Numerical variables were presented as mean and standard deviation (SD), while categorical variables were expressed as frequency and percentage. SEM was used to examine the interrelationship between the variables including direct and indirect relationships. Standardised path coefficient ($\beta$) with 95% Confidence Interval (CI) and $p$-value < 0.050 as significant were determined. The reported output was also assessed for adequacy of fit indices and a significant hypothesised path relationship. The estimator of maximum likelihood robust (MLR) by *Yuan & Bentler (2000)* was used in the SEM analysis. The evaluation of the model's fitness was based on several model fit indices: CFI > 0.92, TLI > 0.92, RMSEA < 0.08, and SRMR ≤ 0.08 (*Hair et al., 2014*; *Kline, 2011*). If inadequate fit indices, the model would re-specified by removing insignificant paths, adding additional correlation between the variance of items based on suggestions from modification indices and inspection of additional possible pathways. All re-specification models were done after adequate theoretical support was considered by the researchers including discussion with the experts.

The models were reviewed after each amendment was done to the model as any changes would bring some effects to the model. The process stopped after a good fit model was achieved. The final and best fit of the structural model to the data was reported at the end of the analysis. The significant paths were concluded with the fit indices as evidence for the validity of the model.

# RESULTS

## Participants

Of 350 screened participants, 323 participants were analysed, while the rest did not satisfy the eligibility criteria. The participants had a mean age of 27.69 years old (SD = 11.50), more than half were males (59.4%), mean BMI was 24.90 (SD = 14.20), half were from rural (52.6%), and 20.4% had other symptoms including headache, nausea, abdominal pain.

## Structural model

The initial structural model has 21 hypothesised path relationships based on the recommendation of the correlation analysis. Table 1 shows the test findings for model 1 to

**Table 1 Findings of fitness tests for model 1 to model 4 ($n = 323$).**

| Model | RMSEA (90%CI) | RMSEA $p$-value | CFI | TLI | SRMR |
|---|---|---|---|---|---|
| Model 1 | 0.060 [0.057–0.063] | <0.001 | 0.757 | 0.742 | 0.094 |
| Model 2 | 0.079 [0.074–0.084] | <0.001 | 0.761 | 0.735 | 0.131 |
| Model 3 | 0.069 [0.063–0.075] | <0.001 | 0.846 | 0.827 | 0.139 |
| Model 4 | 0.048 [0.041–0.054] | 0.714 | 0.922 | 0.908 | 0.064 |

Note:
RMSEA, Root mean square error of approximation; RMSEA $p$-value, Probability of RMSEA ≤ 0.05; SRMR, standardised Root Means Square Residual; TLI, Tucker Lewis Index; CFI, Comparative Fit Index., Model 1-original model with all observed and latent variables based on hypothesis, Model 2-model with two parcel domain for health belief and health-promoting behaviour, Model 3-model with only significant paths (all insignificant path were removed), Model 4-final model with only significant path (hypothesised and additional path) and additional correlation between residual.

model 4 based on fit indices. There were improvements in all the fit indices in the final model, where most of the fit indices had achieved the required thresholds.

The results showed that the final hypothesised model explained the variance of SevGen 6%, depression 9%, physical activity 10%, Sev24 10%, health and treatment awareness 12%, social support 16%, diet 24%, intention 39%, and QoL 53.8%. The detailed description for each significant path is summarised in Table 2. Attitudes were positively associated with intention, social support, and physical activity, whereas they were negatively associated with severity 24 h. Perceived behavioural control, which is also a domain for health belief significantly associated with intention, diet, health and treatment awareness and social support. Intention was positively associated with diet but negatively with severity general and depression. While social support is negatively associated with severity general, severity general significantly increases the impact on QoL. Lastly, stress management negatively impacted depression levels, and depression increased directly with severity 24 h.

The final model 4 is illustrated in Fig. 2.

## Structural model testing for indirect relationship

Two variables (attitude and perceived behavioural control) exhibited positive indirect effects on other outcome variables (severity 24 h, diet under health-promoting behaviour). Table 3 presents the results of indirect relationships in the final model. Attitude had a significant indirect effect on severity 24 h through intention and depression. Perceived behavioural control had a significant indirect effect on diet through intention.

## DISCUSSION

Based on the findings of the current study, health beliefs (including attitude and perceived behavioural control) exhibited a significant direct relationship with intention, a trend observed in various health conditions such as cardiovascular disease, diabetes mellitus, breast cancer, and obesity (*Lim et al., 2021*; *Kueh, Morris & Ismail, 2016*; *Chin & Mansori, 2019*; *Moeini et al., 2017*; *Saghafi-Asl, Aliasgharzadeh & Asghari-Jafarabadi, 2020*). Moreover, intention was found to directly impact diet, a key component of health-promoting behaviour, aligning with findings from other studies (*Kueh, Morris & Ismail, 2016*; *Li, Figg & Schüz, 2019*; *Uchendu, Windle & Blake, 2020*). *Trinh et al. (2012)* suggested that intention plays a crucial role in affecting QoL through detailed planning for

**Table 2 Paths relationship of the final model (Model 4, $n$ = 323).**

| Relationship | Standardised regression coefficient, β (95%CI) | CR | SE | $p$-value |
|---|---|---|---|---|
| ATT→I1 | 0.25 [0.01–0.40] | 3.18 | 0.08 | 0.001 |
| PBC→I1 | 0.42 [0.28–0.57] | 5.68 | 0.07 | <0.001 |
| I1→DIET | 0.13 [0.01–0.25] | 2.10 | 0.06 | 0.036 |
| SS→SEVG | −0.21 [−0.36 to −0.06] | −2.78 | 0.08 | 0.005 |
| SEVG→QOL | 0.73 [0.66–0.81] | 19.87 | 0.04 | <0.001 |
| PBC→DIET | 0.41 [0.28–0.53] | 6.30 | 0.07 | <0.001 |
| PBC→HA | 0.35 [0.22–0.48] | 5.22 | 0.07 | <0.001 |
| ATT→SS | 0.21 [0.01–0.41] | 2.06 | 0.10 | 0.039 |
| PBC→SS | 0.22 [0.03–0.42] | 2.25 | 0.10 | 0.025 |
| ATT→PA | 0.32 [0.16–0.49] | 3.91 | 0.08 | <0.001 |
| ATT→SEV24 | −0.26 [−0.36 to −0.17] | −5.31 | 0.05 | <0.001 |
| I1→SEVG | −0.11 [−0.22 to −0.01] | −2.11 | 0.05 | 0.035 |
| I1→DEP | −0.13 [−0.23 to −0.03] | −2.48 | 0.05 | 0.013 |
| SM→DEP | −0.28 [−0.53 to −0.02] | −2.15 | 0.13 | 0.032 |
| DEP→SEV24 | 0.15 [0.08–0.22] | 3.97 | 0.04 | <0.001 |

**Note:**
SS, social support; SEVG, severity general; QoL, quality of life; I1, intention; DEP, depression; PBC, perceived behavioural control; ATT, attitude; DIET, dietary habit; SM, stress management; ANX, anxiety; PA, physical activity; HA, health and treatment awareness; CR, critical ratios; SE, standard error; β, standardized regression coefficient.

adopting a healthier lifestyle, particularly noted in kidney cancer survivors, emphasizing the importance of intention formation in creating actionable plans. Additionally, perceived behavioural control was shown to influence diet and health outcomes through treatment awareness. *Louis, Chan & Greenbaum (2009)* proposed that perceived behavioural control may directly or indirectly impact behaviour, with intentions playing a mediating role by reflecting actual controls that enhance an individual's capacity to act on their intentions. Various aspects of health behaviour, including eating habits, exercise routines, and health awareness, have been shown to benefit from the TPB, as it provides a framework for predicting dietary patterns and promoting positive health-related actions (*Louis, Chan & Greenbaum, 2009*; *Wang & Wang, 2015*). *Trinh et al. (2012)* introduced health belief as a latent variable within the TPB, suggesting its influence on intention, health behaviors (such as physical activity), and QoL (*Päivi, Mirja & Terttu, 2010*; *Wasshenova, 2018*; *Villoria et al., 2006*; *Orji, Vassileva & Mandryk, 2012*; *Si et al., 2019*). Positive changes in participants' beliefs and intentions regarding AB management have the potential to enhance decision-making processes geared towards positively modifying behaviors to achieve the goal of improved symptom severity and reduced impact on QoL.

The current study highlights a positive correlation between health beliefs (including attitude and perceived behavioural control) and social support. This underscores the interconnected nature of health beliefs in addressing AB through fostering the right attitude, building confidence or control over actions, and cultivating strong social support to achieve the desired intentions or behavioral changes (*Abdullah et al., 2020*; *Lackner et al., 2010*). The significance of belief systems and social support has been widely

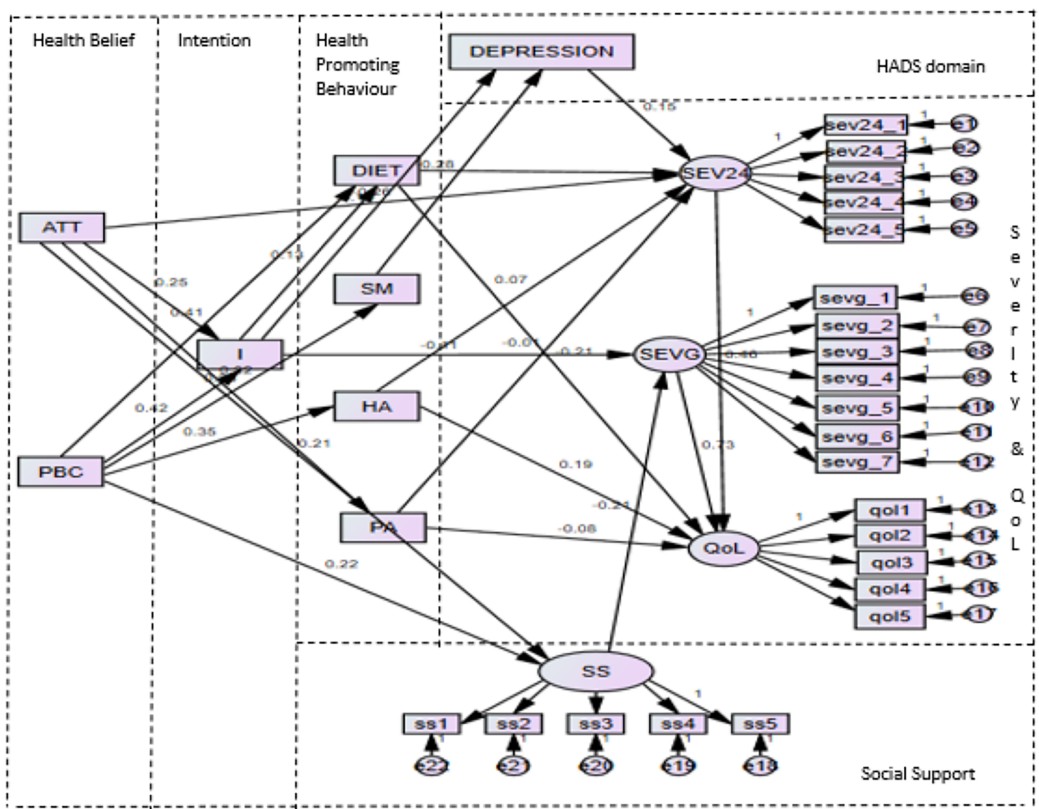

**Figure 2 Finalised SEM model (Model 4).** Note: I, intention; ATT, attitude and PBC, perceived behavioural control from HB-Bloat; DIET, diet; HA, health and treatment awareness; and PA, physical activity from HPB-Bloat; SS, social support from SS-Bloat; DEPRES-SION, depression from HADS; SEVG, severity general and SEV24, severity 24 h from BSQ-M; QoL, QoL from BLQoL-M.

**Table 3 Standardised direct, total indirect, and total effects (n = 323).**

| Predictor variable | | Outcome variable | Causal effect | | |
|---|---|---|---|---|---|
| | | | Direct β (*p*-value) | Indirect β (*p*-value) | Total β (*p*-value) |
| Attitude (under health belief) | → *Via* intention *Via* depression | Severity 24 h | −0.26 (<0.001) | −0.010 (0.034) | −0.27 (<0.001) |
| Perceived behavioural control. (under health belief) | → *Via* intention | Diet | 0.41 (<0.001) | 0.05 (0.030) | 0.46 (<0.001) |

**Note:**
*\**p*-value < 0.05; *Via*, through.

recognised in the literature, demonstrating improvements in health symptoms among different populations, such as older women, individuals with osteoporosis, those with eating disorders, diabetes, fatigue symptoms in multiple sclerosis, and mental health issues (*Hirai et al., 2020*; *Wang et al., 2018*). Research findings indicate that social support plays a pivotal role in influencing QoL through intentions and health beliefs (*Fong et al., 2018*). Furthermore, studies have shown direct relationships between intention and various
factors like socio-demographics, accessibility, and emotional aspects toward social support (*Lo, Guo & Bradley, 2018*; *Jones, Greenberg & Crowley, 2015*). *Lackner et al. (2010)* also noted that patients with IBS who perceive low levels of social support tend to struggle with disease management and exhibit reduced intentions for self-management.

In the final structural equation modeling (SEM) analysis, a significant pathway between intention and overall AB severity was identified, with intention inversely related to general severity levels. This inverse relationship between intention and severity is not extensively documented in the literature, with only a few studies reporting similar findings (*Kim et al., 2019*; *Riaz & Khan, 2016*). Health belief is consistently recognised as a crucial factor in influencing health-seeking behaviour (*Prins et al., 2008*). *Zhang (2017)* observed a more substantial reduction in anxiety and depression scores among participants in the health belief model (HBM)-guided rehabilitation group compared to the conventional group. This highlights the potential of positive health beliefs to stimulate intentions for lifestyle changes that can effectively mitigate the impact of negative emotions, ultimately contributing to reduced severity and enhanced QoL.

The final model of the study revealed a reciprocal relationship between social support and general severity levels. It is hypothesised that individuals with robust social support networks possess the resilience and resources to overcome challenges effectively (*Aghaei et al., 2016*). Additionally, *Lackner et al. (2010)* suggested an inverse association between social support and severity, indicating that higher levels of social support were linked to reduced pain and perceived stress levels among patients with IBS. These findings resonate with similar results from previous studies (*Lackner et al., 2010*; *Aghaei et al., 2016*). General severity emerged as the primary factor influencing QoL, consistent with findings from multiple studies (*Abdullah et al., 2021a*; *Tuteja et al., 2008*; *Malagelada, Accarino & Azpiroz, 2017*; *Lacy & Patel, 2017*; *Kanazawa et al., 2016*). For instance, *Tuteja et al. (2008)* and *Malagelada, Accarino & Azpiroz (2017)* reported a positive correlation between symptom severity and QoL in individuals with AB associated with functional gastrointestinal disorders. Similarly, *Lacy & Patel (2017)* postulated that heightened severity of AB leads to a decrease in QoL and an increase in healthcare utilisation. These findings underscore the critical role of social support and symptom severity in influencing individuals' well-being and health outcomes. The reciprocal relationship between social support and severity, coupled with the substantial impact of general severity on QoL, highlights the interconnected nature of these factors in the context of managing and addressing AB.

The current study conducted among the hospital-community-based adult population in Kelantan aimed to comprehensively explore AB and successfully addressed all objectives regarding the interrelationships among the pertinent variables. The significant findings unearthed in this study provide valuable insights into the correlations among the variables under scrutiny. This knowledge offers a foundation for devising strategies to enhance QoL through various pathways, as illuminated by the study's results. For instance, rather than solely focusing on the impact of severity on QoL, the study suggests that fostering robust social support, enhancing health management practices, nurturing a positive attitude towards self-care coupled with resolute intentions, could potentially mitigate the negative

effects of severity on QoL. These findings underscore the potential for intervention studies aimed at improving the management of AB, fostering healthier lifestyles, and boosting productivity. By increasing awareness of the implications of AB and exploring innovative technologies that can facilitate the treatment of AB in daily routines, there is an opportunity to enhance the well-being of individuals not only dealing with AB but also various other health conditions commonly discussed. It is crucial to disseminate accurate and beneficial information pertaining to AB to diverse audiences and instill positive beliefs to drive intentions for AB-improvement. However, it is essential to acknowledge the limitations of the study. Firstly, the recruitment of adult participants from a specific region within one country may limit the generalisability of the findings to the broader population. Additionally, the accuracy of the data analysis hinges on the thorough completion of questionnaires by the respondents, signifying a potential source of bias or error. For future research endeavors, exploring the effects of guided imagery could be beneficial in fostering psychological well-being, promoting healthy behaviors, and enhancing QoL among individuals grappling with AB. These interventions may contribute to increased productivity and overall well-being of individuals dealing with AB-related challenges. By delving into the potential benefits of guided imagery and its impact on psychological health, behaviour modification, and QoL improvement for individuals coping with AB, future studies can further enrich our understanding and pave the way for tailored interventions to address the multifaceted aspects of AB management.

## CONCLUSIONS

The final SEM analysis revealed 15 statistically significant paths. Among these, direct effects were observed between TPB constructs, SS-bloat, HADS, BSQ-M, and BLQoL-M. Additionally, there were indirect effects noted between TPB constructs and behaviors, as well as symptom severity. These findings hold potential implications for health educators and healthcare providers, particularly in populations affected by AB, notably in Malaysia, aiming to enhance AB management. The empirical evidence generated from these significant paths underscores the importance of healthcare providers in implementing strategies to improve AB among individuals experiencing symptoms. These insights suggest the necessity for interventions and support systems to address AB effectively.

## ACKNOWLEDGEMENTS

We want to thank all the participants who volunteered and participated in the present study. We also want to convey our sincere gratitude to the staff in HUSM for their support and co-operation during the data collection.

### Funding

This work was supported by a RUI Grant of Universiti Sains Malaysia: 1001.PPSP.8012250. The funders had no role in study design, data collection and analysis, decision to publish, or preparation of the manuscript.

## Grant Disclosures

The following grant information was disclosed by the authors:
RUI Grant of Universiti Sains Malaysia: 1001.PPSP.8012250.

## Competing Interests

Yeong Yeh Lee is an Academic Editor for PeerJ.

## Author Contributions

- Nurzulaikha Abdullah conceived and designed the experiments, performed the experiments, analyzed the data, prepared figures and/or tables, authored or reviewed drafts of the article, and approved the final draft.
- Yee Cheng Kueh conceived and designed the experiments, performed the experiments, analyzed the data, prepared figures and/or tables, authored or reviewed drafts of the article, and approved the final draft.
- Garry Kuan conceived and designed the experiments, performed the experiments, authored or reviewed drafts of the article, and approved the final draft.
- Mung Seong Wong conceived and designed the experiments, performed the experiments, authored or reviewed drafts of the article, and approved the final draft.
- Vincent Tee conceived and designed the experiments, authored or reviewed drafts of the article, and approved the final draft.
- Tengku Ahmad Iskandar Tengku Alang conceived and designed the experiments, authored or reviewed drafts of the article, and approved the final draft.
- Nurhazwani Hamid conceived and designed the experiments, authored or reviewed drafts of the article, and approved the final draft.
- Yeong Yeh Lee conceived and designed the experiments, performed the experiments, prepared figures and/or tables, authored or reviewed drafts of the article, and approved the final draft.

## Human Ethics

The following information was supplied relating to ethical approvals (*i.e.*, approving body and any reference numbers):

Ethical approval from the Human Research Ethics Committee of Universiti Sains Malaysia (USM/JEPeM/17010012) was obtained.

## Data Availability

The data is available in the Supplemental File.

## Supplemental Information

Supplemental information for this article can be found online at http://dx.doi.org/10.7717/peerj.17265#supplemental-information.

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
