# Peer review of "Structural equation models of health behaviour, psychological well-being, symptom severity and quality of life in abdominal bloating"

_PeerJ, doi:10.7717/peerj.17265_

## Round 0.1 · original submission · Major Revisions

I have now received the reviewers' comments on your manuscript. They have suggested some major revisions to your manuscript. Therefore, I invite you to respond to the reviewers' comments and revise your manuscript.

**Language Note:** The review process has identified that the English language must be improved. PeerJ can provide language editing services - please contact us at copyediting@peerj.com for pricing (be sure to provide your manuscript number and title). Alternatively, you should make your own arrangements to improve the language quality and provide details in your response letter. – PeerJ Staff

Reviewer 1 ·

Basic reporting

Structure:
Regarding the structure of the manuscripts there are a few abbreviations which were not introduced at the first mention, e.g. IBS, IBD and GI in lines 49, 56, and 109, respecively. There is a inconsistent use abbreviations of the "Rome IV" (line 103) and "ROME criteria (line 108). Furthermore, the authors are using inconsistent referencing styles, e.g. line 174 and 214-215. The resolution of the figures could be higher. Especially figure 2, which contains a lot of details, is difficult to read.

Language:
As far as the language is concerned, there are quite a few mistakes. I have tried to add suggestions for improvement, but would recommend having the entire article proofread by a fluent English speaker.
- lines 32-33: “A total of 323 participants with a mean age of 27.69 years old (SD = 11.50) [insert e.g.: participated in the study”], and the majority were female (64.7%).”
- line 46: “According to Tuteja et al. (2008); [“,” instead of “;”] AB was one of the…”
- line 51-53: “Even though AB symptoms were [“are”] not life threatening, several researchers reported it as a cause for the limitation in daily activities (Kanazawa et al., 2016)”.
- lines 61-63: “Some literature suggests that IBS significantly lowers the QoL and AB, which are [“is”] commonly present in people with IBS and other diseases was [“is”] no different (Kanazawa et al., 2016; Thiwan, 2019; Kamboj & Oxentenko, 2018).”
- lines 64-66: “Therefore, more research was initiated to explore the alternative ways to treat AB, especially from all aspects, not only from a clinical point of view but also from the in-fluence of culture or environment.” I don’t really understand what cultural or environmental variables you are referring to. If you mean the psychological / social variables you later specify in your model, I think the terms “culture” or “environment” are not accurate.
- lines 67 – 68: “For nearly two decades, other than [“besides”] medication, other psychological or social variables have been considered a foundation of AB management (Drossman, 2006; Tanaka et al., 2011).”
- lines 70-73: “Psychologically, the Theory of Planned Behaviour (TPB) proposed that belief, intention, and be-haviour [“behaviour”] correlate to each other and manipulating these three cores concept [“core concepts”] may help to improve a symptom or disease (Ajzen, 2012).”
- lines 77-78: “Symptom of AB poses a disturbance in almost any individual’s busy daily routine as it dramatically [avoid unscientific language] impacts one’s well-being.”
- lines 78-79: “Besides, it might also trigger other problems and deteriorate health, thus it should not be taken lightly [avoid unscientific language].”
- lines 79: “Yet, the researchers found a lack of research [unprofessional language] on the related variables among Malaysian people with AB.”
- line 81-82: “Hence, the study aims to determine the direct and indirect relationships between TPB psychological construct [psychological constructs of the TPB], …”
- line 136: “…5-point Likert scales [a 5-point Likert scale]…”
- line 141: “…[missing “(“] Abdullah et al.,2021)…
- lines 176-178: “For SevGen, items 1, 3,4,5, and 6 measured 1-5 scale [used a scale ranging from 1 to 5], item 2 measured 1-4 scale [was measured on a scale ranging from 1 to 4], and item 7 measured 1-7 scale [was measured on a scale ranging from 1 to 7]. For sev24, all items [were] measured 1-5 scale [on a scale from 1 to 5] except for item 5 (estimated [what do you mean by “estimated”?] 1-8 scale).”
- line 178 – 179: “…(Cronbach alpha values for SevGen and Sev24 were 0.76 and 0.85, respectively (Palsson et al., 2004)[“)”].”
- line 186: “…relation-ships [“relationships”]…”
- line 194-195: “The study was conducted around [“the”] Hospital Universiti Sains Malaysia, Kelantan, Malaysia where [“a”] purposive sampling method was employed.”
- line 244: “…, which is also a domain for health belief [“was”] significantly…”
- line 322: “…ac-curate [“accurate”]…”

Background context:
In lines 73-76 the authors write: “Initially developed in 1990 by Ajzen, the TPB evolved through the ideas which circled around the prediction of behaviour and extended originally from the Theory of Reasoned Action (TRA) with the additional domain of perceived behaviour control under the health belief construct (Ajzen, 2012).”
The fact that TPB evolved from TRA is redundant information in the context of this study. Furthermore, I did not really get the main points of the TPB from your explanations. Since the TPB forms the theoretical backbone of your model, I would make sure that you first describe the TPB in a precise and understandable general way and then relate it to the context of health-promoting behaviour in abdominal bloating.

Experimental design

Research question:
I strongly recommend that you go into more detail about the individual components of the model (figure 1) in the introduction and justify in terms of content why and how exactly these components are related to each other. Cite sources that support the assumed relationship between the model components. Apart from the components of the TPB, the model otherwise seems somewhat arbitrary.
Additionally, I recommend further elaborating on why and how an understanding of the relationship among the model components can positively influence patients' symptom management. I don't think the knowledge gap is well defined. Furthermore, how your study intends to close this gap should be more clearly elaborated.
For example, in lines 80-81 you remain quite superficial with stating: “The application and exposure of these constructs to Malaysian with AB could positively affect their symptom management.“. The question remains, how exactly a better understanding of the relationship between the components of your model could positively affect symptom management. You continue and conclude: “Hence, the study aims to determine the direct and indirect relationships between TPB psychological construct, other psychosocial variables, symptom severity, and QoL among Malaysian with AB using structural equation modelling (SEM).” I don't really see a well-reasoned explanation as to why you assume the proposed relationship between the model components.

Lack of detail in Methods section:
- lines 124-125: This section could benefit from the specification of sample items for each subscale.
- line 160: “The 14-item HADS is used to assess the presence of psychological disorders, …”. The HADS is a screening instrument, which is not sufficient to assess the presence of any psychological disorder.
- line 179-181: “There is an abbreviated 3-item Sev24, with similar internal consistency to 5-item Sev24, but the original 5-items Sev24 was used (Palsson et al., 2004; Thiwan et al., 2005; Thiwan, 2004).” Unnecessary information in the context of your study.
- line 196: How exactly were participants screened regarding the presence of mental disorders?
- line 233: “The initial structural model has 21 hypothesised path relationships.” This number of path relationships does not emerge from the model postulated in the introduction (Figure 1). On what theoretical basis was this model specified?

Validity of the findings

Conclusions:
The discussion remains descriptive and reflects some of the shortcomings of the introduction (e.g., lack of a well-defined knowledge gap). The significant findings are reported and placed in the existing literature. However, the implications of the results remain almost entirely undiscussed. Regarding the implications of the findings, there is only one statement in lines 328-329: “These findings could benefit health educators and health care providers in improving the management of AB.” How exactly these results can contribute to improving AB management remains unclear to me.

Reviewer 2 ·

Basic reporting

The language of the manuscript needs careful editing as it contains numerous errors. The entire text requires thorough revision to make it more fluent. For example, there are errors in the sentence in the abstract: "A total of 323 participants with a mean age of 27.69 years old (SD = 11.50), and the majority were female (64.7%)." Some places lack parentheses, while others have an extra one. In line 58, it should be "despite" instead of "de-spite." Similar errors can also be found elsewhere.

Abbreviations should be written in full when they first appear. For example, what is the full name of IBS?

The titles of the figures and tables are not standardized.

Experimental design

In the Introduction section, the author dedicates a significant portion to introducing the harms of AB, but provides a very brief overview of TPB and AB research background.

The author's train of thought in the introduction is unclear. After interweaving the introduction of AB's current status and influencing factors, the author finally introduces the purpose of the study - the association between TPB and AB severity. However, the author suddenly starts discussing the symptoms of AB and their impact on people's daily lives, as well as introducing other theoretical models.

For the inclusion of social support, depression, and anxiety as variables in the model, the author did not provide a reasonable and sufficient explanation in the Introduction.

Regarding the questionnaires used in the study, please provide information on the reliability and validity within the study population.

Can depression and anxiety be diagnosed solely based on the HADS questionnaire? I believe these are symptoms.

Did the author consider the influence of other covariates in the analysis, such as socioeconomic status, medical history, family history, etc.?

Why were only AB patients included and not a control group of healthy individuals? How did the author consider this?

Validity of the findings

no comment

Additional comments

no comment

---

## Round 0.2 · accepted · Accept

The authors have made all the changes. I have no further comments.